Development of a camera trap for perching dragonflies: a new tool for freshwater environmental assessment

Yoshioka Akira yoshioka.akira@nies.go.jp 1
Shimizu Akira 2
Oguma Hiroyuki 3
Kumada Nao 3
Fukasawa Keita 3
Jingu Shoma 4
Kadoya Taku 3
1 Fukushima branch, National Institute for Environmental Studies , Miharu , Fukushima Prefecture , Japan
2 Ami , Ibaraki Prefecture , Japan
3 Center for Environmental Biology and Ecosystem Studies, National Institute for Environmental Studies , Tsukuba , Ibaraki Prefecture , Japan
4 Faculty of Human Sciences, Waseda University , Tokorozawa , Japan
Lambert Max
Electronic publication date: 2020 Sep 18
Publication date: 2020
Volume: 8
Electronic Location ID: e9681
Received 2020 Mar 30; Accepted 2020 Jul 17
Copyright: ©2020 Yoshioka et al.
Copyright year: 2020
Copyright holder: Yoshioka et al.
License: This is an open access article distributed under the terms of the Creative Commons Attribution License, which permits unrestricted use, distribution, reproduction and adaptation in any medium and for any purpose provided that it is properly attributed. For attribution, the original author(s), title, publication source (PeerJ) and either DOI or URL of the article must be cited.
License URL: https://creativecommons.org/licenses/by/4.0/

Keywords: Autodetection, Biodiversity, Camera trapping, Ecological monitoring, Odonata

Funding: Japan Society for the Promotion of Science 16H05061 18K05931 This study was funded by Japan Society for the Promotion of Science (grant ID: 16H05061 and 18K05931). The funders had no role in study design, data collection and analysis, decision to publish, or preparation of the manuscript.

==============================
Although dragonflies are excellent environmental indicators for monitoring terrestrial water ecosystems, automatic monitoring techniques using digital tools are limited. We designed a novel camera trapping system with an original dragonfly detector based on the hypothesis that perching dragonflies can be automatically detected using inexpensive and energy-saving photosensors built in a perch-like structure. A trial version of the camera trap was developed and evaluated in a case study targeting red dragonflies (Sympetrum spp.) in Japan. During an approximately 2-month period, the detector successfully detected Sympetrum dragonflies while using extremely low power consumption (less than 5 mW). Furthermore, a short-term field experiment using time-lapse cameras for validation at three locations indicated that the detection accuracy was sufficient for practical applications. The frequency of false positive detection ranged from 17 to 51 over an approximately 2-day period. The detection sensitivities were 0.67 and 1.0 at two locations, where a time-lapse camera confirmed that Sympetrum dragonflies perched on the trap more than once. However, the correspondence between the detection frequency by the camera trap and the abundance of Sympetrum dragonflies determined by field observations conducted in parallel was low when the dragonfly density was relatively high. Despite the potential for improvements in our camera trap and its application to the quantitative monitoring of dragonflies, the low cost and low power consumption of the detector make it a promising tool.

Introduction

Recent advances in computing power, digital cameras, batteries, acoustic devices, and other digital tools have facilitated the development of novel techniques for long-term and broad-scale biodiversity observations, which are required for successful large-scale conservation and planning (Bush et al., 2017). In particular, camera traps and unmanned aerial vehicles have made it possible to effectively investigate wildlife populations (Rowcliffe et al., 2008; Hodgson et al., 2016; Nakashima, Fukasawa & Samejima, 2018). Methods for the acoustic monitoring of birds and frogs are also rapidly advancing (Acevedo & Villanueva-Rivera, 2006).

Many environmental indicator organisms, however, have not been adequately monitored using digital tools. Dragonflies (Odonata) are a prominent example. They are often assumed to be important indicators for assessing freshwater ecosystems (Clark & Samways, 1996; Kadoya, Suda & Washitani, 2004; Yoshioka et al., 2014). Furthermore, as multi-habitat dwellers, they can indicate the status of complex aquatic and terrestrial ecosystems, including traditional agricultural landscapes, which support considerable biodiversity (Kadoya et al., 2008; Kadoya & Washitani, 2011; Yoshioka et al., 2017). However, existing survey methods for dragonflies are mainly based on field observations by experts or researchers (e.g., Kadoya et al., 2008). This limits the area-wide monitoring of dragonflies because field observation methods are sensitive to the skill of investigators and to weather conditions.

One potential method for monitoring dragonflies is the camera trap technique, a common approach for monitoring mammalian taxa. Commercial camera traps are relatively inexpensive have a high resolution and are easy to maintain owing to their low power consumption; it is only necessary to collect the digital records and change the batteries a few times a year (Fukasawa et al., 2016; Yoshioka, Mishima & Fukasawa, 2016). However, these camera traps usually target homeotherms and are triggered by passive infrared sensors, such as a pyroelectric sensors, which are not usually applicable to insect detection. Some active photoelectric sensors can sensitively detect insects (Kawada & Takagi, 2004; Silva et al., 2015). However, they require a considerable amount of power and are relatively expensive owing to their light projection systems. Thus, the cost to maintain these types of traps becomes prohibitive when used in the field for an adequate period to survey insects.

The behavioral characteristics of dragonflies provide a basis for resolving this issue. In particular, many species of dragonflies are referred to as “perchers” (Corbet, 1999); they often perch on the highest point of a structure to search for prey and competitors (Gorb, 1994; Switzer & Walters, 1999; Kadoya, Suda & Washitani, 2004; Iwasaki, Suda & Watanabe, 2009). We hypothesized that perchers can be attracted to the top of an artificial perch and that the consequent interception of direct solar light at the top of the perch can be detected using passive light sensors, thereby triggering a camera trap. According to this hypothesis, passive light sensors, which are inexpensive and consume minimal electric power, are expected to effectively capture perching insects.

Our aim was to contribute to the general question: can camera traps be used to monitor dragonfly assemblages? In particular, we developed camera traps using passive light sensors for the detection of percher dragonflies. First, we designed camera traps consisting of a commercially available camera and an original dragonfly detection device, based on the idea that perchers can be detected using an artificial perch with passive light sensors. To enable its application to various percher species, the detection algorithm included various tunable parameters. Then, as a case study, we implemented the camera trap for the detection of Sympetrum dragonflies in a short-term field experiment. The detection frequency and sensitivity were evaluated at three sites to obtain basic information on the accuracy of the trap. In addition, data from the camera trap were compared with data from independent observations by an investigator to preliminarily explore whether the trap detection frequency is a good indicator of the population density.

Materials & Methods

Fundamental concept of percher dragonfly detection using passive light sensors

The basic structure of the camera trap for percher dragonfly detection is shown in Fig. 1 (please note that the National Institute for Environmental Studies obtained a national patent for the basic idea (Japan Patent Office, 2019). In the cylindrical rod-like structure (i.e., the detector section), the first light sensor (or photoresistor) is set at the top and the second light sensor is set under the first sensor. These sensors should be separated by a distance that is longer than the body size of a dragonfly. At the top of the cylindrical structure, there is a transparent window above the first sensor. Around the second sensor, the wall of the cylindrical structure consists of a light diffusion material. These sensors are connected to a microcomputer to process the signals. The circuit of the dragonfly detection device is powered by a DC battery.

Figure 1 Schematic diagram of the proposed camera trap for Sympetrum dragonfly detection.

The processor section is connected to a digital camera oriented at the top of the cylindrical structure. The camera is set to be triggered when the signal values from the two sensors meet a threshold condition, and the microcomputer outputs an imaging command.

When a dragonfly perches on the top of the detector, the direct solar radiation being detected by the first sensor is expected to be sufficiently blocked, while diffuse solar light being detected by the second sensor should remain relatively unchanged. Therefore, the relationship between the signals from the two sensors is expected to change, triggering the connected camera when the dragonfly is perching.

Design and materials for the camera trap

Using inexpensive and common materials, we assembled trial camera traps (Fig. 2A; Fig. S1). In the detector section, two cadmium sulfide (CDS) cells with 1 MΩ dark resistance (GL5528; Nanyang Senba Optical & Electronic Co., Ltd.) were used as light sensors. This passive light sensor cost about USD 0.25. The upper cell was covered by an LED silicone cap (OptoSupply Ltd.). The bottom of each cell was colored black using a permanent marker. These cells were connected to a microprocessor via USB cables. To enclose the cells, an acrylic pipe (10 mm outer diameter, seven mm inner diameter, and 500 mm length) was used. The top of the pipe was surrounded by 150-mm-wide masking tape (Sekisui Chemical Co., Ltd.) as a transmission diffuser, as well as 20-mm-wide black vinyl tape, and capped with a 0.5-mm-thick polyethylene terephthalate plate (Acrysunday Co., Ltd.) using Aron Alpha adhesive (Toagosei Co., Ltd.).

Figure 2 (A) A trial camera trap used for the field experiment and (B) an image of Sympetrum infuscatum obtained by the camera trap.

In the processor section, a PIC microcomputer (PIC16F1827-I/P; Microchip Technology Inc.) was used as the main processor. The microprocessor was connected to a > 2.3 V DC battery (three AA batteries were used in the experiment described below). Resistance values for each CDS cell in the detector section, which decrease with illumination intensity, were converted to signal values of 0–1023 and processed using the microcomputer. The processor section was enclosed in a polyvinyl chloride (PVC) pipe. To prevent rainwater intrusion, two silicon caps penetrated by the detector section were inserted on the top of the PVC pipe. This PVC pipe was attached to a tripod to adjust the height of the camera trap. (In the following experiment, the height was about 2 m.)

Figure 3 Schematic algorithm for dragonfly detection.

Each cycle of the process is executed at 1-s intervals.

A commercial digital camera covered in a water-proof case (TLC 200; Brinno Inc.) was connected to the microprocessor using video cables (VX-ML10G or VX-ML20G; JVC Kenwood Co.). Note that the camera is usually used as a time-lapse camera and can function for dozens of days with four AA batteries if controlled appropriately. The camera was attached to the top of the PVC pipe to capture the top of the detector section.

Dragonfly detection algorithm

A simple detection algorithm for a field experiment was developed (Fig. 3). This algorithm was aimed at directing the connected camera to obtain a single image per perching dragonfly. That is, once an image of a dragonfly on the top of the sensor was obtained, the next photo should not be taken until after that dragonfly took off and another one perched on the sensor. In addition, to prevent the trap from detecting instantaneous light from organisms or materials other than dragonflies, a continuous perching signal for a given time period (“Z” in Fig. 3) was required. Issues related to individual differences in the sensitivity of CDS cells and detector sections were minimized by using the ratio of relative signal values (temporal change in the instantaneous ratio of two signal values to the average) as an indicator for dragonfly detection, rather than the ratio of absolute signal values.

Case study

Sympetrum dragonflies

In Japan, dominant Sympetrum species, such as S. frequens, are perchers and are considered environmental indicators for agricultural landscapes consisting of paddy fields (Sprague, 2003; Inoue & Tani, 2010; Tanaka, 2016). Mature adult Sympetrum dragonflies have conspicuous red bodies in the autumn and are familiar to people in Japan (Inoue & Tani, 2010). There are concerns about the rapid decrease of these species in some regions in Japan due to pesticide use (Jinguji & Ueda, 2015). In addition, land abandonment can reduce aquatic ecosystems in Japan (Ikegami, Nishihiro & Washitani, 2011), which can impact these dragonflies.

Considering that the intensification and/or abandonment of paddy ecosystems are common in rice-producing regions, continuous broad-scale monitoring of these dragonflies is needed. However, monitoring methods for these species are fairly limited. Insects are usually observed by experts or trained citizens (Inoue & Tani, 2010; Tanaka, 2016). Therefore, examining the applicability of our camera trap to Sympetrum species has implications for conservation ecology. Considering that the genus is distributed worldwide, except in Oceania and south-central Africa (Sugimura et al., 1999), Sympetrum dragonflies were an ideal candidate for assessing our camera trap.

Detection parameters

In the field experiment, parameters for the detection algorithm were provisionally set based on empirical knowledge. The interval to reset the average ratio of signal values X was set to 300 s, and the threshold value Z for perching duration was set to 10 s. The threshold value for Y, the ratio of the instantaneous relative signal value to the average relative signal value, was set 1.05. The processor commanded the digital camera to reboot when the perching duration Td reached 7 s because the digital camera slept. Note that the microprocessor could not monitor the status of the camera and the reboot signal was similar to that for taking a photo. In addition, the microprocessor was set to periodically wake up after sleeping for 1 s to save power. Considering that three 1.2 V, 1,900 mAh batteries were used per detector and lasted at least 66 days during the preliminary study described below, power consumption was estimated to be less than 5 mW. The algorithm (Code S1) was described and compiled using MPLAB IDE v. 8.60 (Microchip Technology Inc.) and microprocessor programming was performed using the MPLAB ICD3 In-Circuit Debugger (Microchip Technology Inc.).

Field experiment

Study sites and camera trap setting. In a preliminarily experiment, a camera trap was set in the yard of the Fukushima branch, National Institute for Environmental Studies (NIES), Miharu town, Fukushima prefecture, Japan (37.434°N, 140.519°E) from October 11 to December 16, 2016 (Fig. 2A). Note that dragonflies are very sparse in the region after November (A Yoshioka, pers. obs., 2016).

A short-term field experiment was also conducted to examine the performance of the camera trap. From 8:05 on October 21 to 17:30 on October 22, 2016, three camera traps were set in the yard of the NIES, Tsukuba city, Ibaraki prefecture, Japan (36.049°N, 140.117°E; Figs. 4A–4C). The average temperatures on the two days were 14.5 °C and 14.9 °C, respectively, and precipitation was not recorded (Japan Meteorological Agency http://www.data.jma.go.jp/obd/stats/etrn/index.php?prec_no=44&block_no=&year=&month=&day=&view, accessed February 7, 2018). This season corresponds to the late stage of adult dragonflies (Inoue & Tani, 2010). Two camera traps were set in areas where the density of the dragonflies was relatively high, one in a field and one a pond. The field site consisted of small experimental paddy fields neighboring a greenhouse. The paddy fields were apparently not managed in the season and vegetation within the paddies was relatively short. Sympetrum dragonflies rarely perched on the vegetation within paddies. Instead, they perched on artificial poles and net-fences, or trees surrounding the site. The pond site consisted of a small pond and wet vegetation, in addition to paved ground. The Sympetrum dragonflies were frequently observed to perch on cattails (Typha spp.) and tall goldenrods (Solidago canadensis). The remaining trap was set on the rooftop of the main building of the institute, where no vegetation exists. There were some artificial poles, but dragonflies rarely perched at the site, which is relatively distant from aquatic habitats (thus, dragonflies were expected to be rarely detected). Each site was about 250 m2 considering the homogeneity of land cover and accessibility for the dragonfly census described below. In addition, these sites were more than 150 m apart and intercepted by multiple buildings and trees.

Figure 4 Field experiment sites, including (A) field, (B) pond, and (C) rooftop sites, and (D) a photo of a Sympetrum dragonfly obtained by the camera trap at the pond site.

Figure 5 (A) Schema of a “perching event” extracted from a movie captured by the time-lapse camera for validation. (B) Number of perching events automatically detected by the camera trap at each site.

Black bar shows the number of detected perching events. White and grey bars show undetected perching events continuing for more than 20 s and less than 20 s (perhaps less than 10 s), respectively.

Evaluation of auto-detected images. Images obtained from the three camera traps were used to count true positives and false positives (i.e., photos with and without Sympetrum dragonflies). The number of false positives indicated the extent of type I error. Each image was visually evaluated. When there were multiple photos within a few seconds, the first photo was removed from the analysis (see also the “not analyzed to avoid double count” column in Data S1). These images resulted from commands for rebooting the camera when it was already active and led to overestimates of true positives.

Validation by time-lapse camera monitoring. To check perching dragonflies undetected by the camera traps (i.e., false negative or type II errors) and to evaluate the detection sensitivity, an additional digital camera was placed near each camera trap. At the field and pond sites, time-lapse cameras (TLC 200) captured movies at 10-s intervals. At the rooftop site where dragonflies were expected to be rare, a video camera captured movies at 30 fps, instead of a time-lapse camera. After the experiment, the movies were thoroughly checked and the presence or absence of Sympetrum dragonflies in each frame was recorded. Movies were initially used to screen for files that include perching dragonflies. Then, movie files including perching dragonflies were split to jpeg image frames using FFmpeg ver. 2015-04-02 (https://ffmpeg.org/) to evaluate frame numbers. No dragonflies perched on the camera trap at the rooftop site during the experiment. To calculate sensitivity, defining a false-negative case based on movie files obtained from the cameras for validation was required. Because perching dragonflies often showed instantaneous takeoff and landing, a “perching event” was defined as an array of movie frames including fewer than two continuous frames without a dragonfly (Fig. 5A). That is, if a dragonfly left the camera trap for more than two frames (20 s), the perching event was closed. Note that 20 s corresponds to the bout criterion interval (Slater, 1974) and the number of “perching events” did not change substantially by increasing the intervals to longer than 20 s. Based on the timestamps for the image frames, the start and end times of each perching event were obtained. Then, the detection of each perching event by camera traps was verified by matching timestamps of autodetected images. If at least one image capturing a Sympetrum dragonfly was obtained by the camera trap during a perching event period, the perching event was recorded as a true positive. If no image of a dragonfly was detected during an event, this was recorded as a false negative. Then, sensitivity was estimated for each site as (true positives ÷ (true positives + false negatives)). A caveat is that an individual could contribute to multiple perching events, and the replacement of perching individuals within 20 s is possible. Thus, sensitivity for a camera trap may reflect the activity of dragonflies, but not necessarily the population density. In addition, Fisher’s exact test was used to compare the frequencies of true positives and false negatives among sites using the fisher.test function implemented in R version 4.0.0 (R core Team, 2020).

Comparison with the dragonfly census. To examine whether detection by the camera traps reflects the dragonfly population density, a comparison among survey methods is important. In parallel with the short-term experiment, census data for Sympetrum dragonflies were obtained at each site. As a standardized census method, a timed survey (Balzan, 2012; Harabis & Dolny, 2015) was adopted because a transect survey was not suitable for the non-linear configurations of sites. At each site, an investigator counted and recorded the number of Sympetrum dragonflies for 10 min three times a day (six observations in total). Previous studies have reported that marked individuals of some perching dragonfly species could maintain reproductive territories for about 1.5–2.5 h (Koenig & Albano, 1985; Moore & Martin, 2016). Thus, three 3-hour time periods were established (9:00–12:00, 12:00–15:00, and 15:00–18:00) and all of the sites were visited for the census for each time period (note that dragonflies were rarely observed from 8:00 to 9:00). For each time period, the first site visited at first was randomly determined and the remaining two sites were surveyed within 1 h from the start of the first survey (see Data S3 for details). Therefore, the effect of sites was not confounded with variation among time periods. During the timed survey, the investigator slowly and randomly walked while searching for dragonflies within the site, because the dragonfly density was relatively low and counting individuals at fixed points was not likely to be effective. To minimize double counting, perching individuals at the same point were counted only once for each 10-min survey, unless the investigator observed individuals leaving the point. For each site, the numbers of Sympetrum dragonflies observed per 10 min period were summed to obtain the “total frequency of dragonflies observed”.

To obtain robust inferences on the relationship between camera detection and census observations given the small sample size, two statistical analyses were used (see Code S2). Note that the number of images of Sympetrum dragonflies was used as an index for camera detection rather than the perching events owing to the low values at the field and rooftop sites (one and zero, respectively). First, differences in Sympetrum dragonfly detection by the camera trap among sites and differences in the total frequency of dragonflies observed among sites were evaluated. If camera detection indicates the dragonfly population density, similar to census observation, statistical inferences should be similar for the two approaches. To test the differences among the sites, a goodness-of-fit test was applied to each pair of sites under the null hypothesis that the probability of dragonfly occurrence is equal between sites (i.e., 0.5). The probabilities of obtaining data (or a more extreme difference than the observed value) under the null hypothesis were computed by Monte Carlo simulations (Hope, 1968) with 1,000,000 replicates using the chisq.test function implemented in R version 4.0.0 (R Core Team, 2020). The random seed was set to 0. The significance probabilities were adjusted by Holm’s method (Holm, 1979) to account for multiple comparisons. Next, a direct approach was used to examine relationships between camera detection and census observation. Because the site number was small, the sample unit was set to dragonfly detection over a 3-hour time period per day per site and census observation corresponding to the time period (i.e., n = 18). This assumption was based on the previous studies that perching individuals are usually replaced within 3 h (Koenig & Albano, 1985; Moore & Martin, 2016). Then, a generalized linear mixed model (GLMM) with a quasi-Poisson error distribution was used to evaluate whether the number of dragonflies detected per 3-hour period (the response variable) reflected the number of dragonflies observed per 10 min (the explanatory variable). In the model, site, day, and time period were included as random variables. The GLMM was implemented using the glmmTMB function in the glmmTMB package for R version 4.0.0 (Brooks et al., 2017; R Core Team, 2020). Similarly, the GLMM was calculated using the number of dragonflies detected as the explanatory variable and the number of dragonflies observed as the response variable. The caveat is that the adherence of an individual Sympetrum dragonfly (mainly S. frequens) to a territory is unclear. Nevertheless, the territorial behavior of S. frequens is not remarkable (Sugimura et al., 1999) and their adherence to a specific perch may be relatively weak. In this study, the only individual that perched on the camera trap in the field site held its territory for 25 min (from 13:30 to 13:55 on October 22) at most (see Data S2). At the pond site, an individual was considered to hold its territory for 2.4 h (from 12:08 to 14:32 on 21th October) at most, if we assumed that the replacement of individuals was indicated by leaving the perch for more than 180 frames (30 min), which is an adequate period of time for copulation and tandem oviposition (Sugimura et al., 1999; Ishizawa, 2012). Thus, a 3-hour time period should be sufficient for the replacement of perching individuals.

Results

Successful detection of Sympetrum species

Through the preliminary experiment and the short-term experiment, we confirmed that the developed camera traps were able to detect Sympetrum dragonflies in the outdoor environment. As expected, our camera traps in the field experiments automatically obtained images of Sympetrum dragonflies (Figs. 2B, 4D).

Figure 6 Number of images automatically obtained by the camera trap at each site.

Black bar and grey bars show true positives (photos with a perching dragonfly) and false positives (photos without dragonflies), respectively.

During the preliminary experiment with a single camera trap, the batteries for the processor section (three AA nickel-metal hydride batteries) were not exhausted during the study period. However, the batteries for the connected camera (four AA alkaline batteries) were exhausted after November 5th, exchanged on November 8th, and exhausted again after December 12th. Among 1,609 images, seven captured S. infuscatum (Fig. 2B).

During the short-term experiment using three camera traps, we detected S. frequens and S. darwinianum. Note that although we could identify S. infuscatum to the species level from the images based on the conspicuous dark color of the tips of wings, we could not distinguish S. frequens from S. darwinianum owing to the low resolution. Furthermore, during the case study, no images detecting other flying organisms were obtained (i.e., the observed false positives could be explained by external abiotic factors, such as the movement of the sun and clouds, or to unexpected internal device errors).

Frequency of successful detection and type I errors

To quantify to what extent the automatic detection correctly captured Sympetrum dragonflies, the images obtained from the camera traps in the short-term experiment were checked. The frequency of true positives was highest at the pond site (50 pictures) and was lowest at the rooftop site (zero pictures) (Fig. 6; Data S1). The numbers of false positives were 49, 51, 17 at the field, pond, and rooftop sites, respectively.

Frequency of type II errors and sensitivity

Then, the extent to which the camera traps sensitively detected perching Sympetrum dragonflies without overlooking was quantified using data from the camera traps and the time-laps cameras. The numbers of true positives on perching events were 1, 8, and 0 at the field, pond, and rooftop, respectively, while the numbers of false negatives were 0, 4, and 0, respectively. Most perching events longer than 10 s could be detected (Fig. 5B; Data S2). The sensitivity values based on perching events were 1.0 and 0.67 for the field and the pond sites, respectively. Although all false negatives were recorded at the pond site, a one-sided Fisher’s exact test (p = 0.6923) did not indicate that the ratio of false negatives at the pond site was higher than that at the field site owing to the small sample size. Note that the rooftop site was not analyzed owing to the lack of perching events.

Comparison with census data

Simultaneously, the census of the dragonflies was conducted by an investigator to be compared with the result of camera trapping. The total frequency of Sympetrum dragonflies observed was highest at the pond site (35 individuals) and lowest at the rooftop site (8 individuals) (Fig. 7; Data S3).

Figure 7 (A) Number of images of Sympetrum dragonflies detected by the camera trap at each site and (B) total frequency of Sympetrum dragonflies observed by the investigator at each site.

The “total frequency” corresponds to the total number of Sympetrum dragonflies observed during six times 10-min surveys. Bars with the same lowercase letters in each graph were not statistically different (A goodness-of-fit test was applied to each pair of sites). Note that the graph in (A) is same as the graph of true positives in Fig. 6.

The goodness-of-fit tests on the detection frequency by camera traps showed that there was a statistically significant difference (significance level of adjusted p-value was 0.05) among the three sites (Fig. 7). The adjusted p-values based on Holm’s method for pairwise comparisons were 0.000036 (field vs. pond), 0.000003 (pond vs. rooftop), and 0.000074 (field vs. rooftop). For investigator observations, we did not obtain a significant difference between the field and pond sites, with adjusted p-values based on Holm’s method of 0.0814 (field vs. pond), 0.0486 (pond vs. rooftop), and 0.000129 (field vs. rooftop).

The GLMM showed that the effect of census observation per 10 min period on camera detection for a 3-hour period was significantly positive (estimated regression coefficient  ± S.E. = 0.4040  ± 0.2024, Z-value = 1.996, p-value based on the Z-value = 0.0459). However, in the model examining the effect of camera detection on census observations (i.e., in which explanatory and response variables were replaced with each other), the regression coefficient was not significant (estimated regression coefficient ±  S. E = 0.02764  ± 0.01726, Z-value = 1.602, p-value based on the Z value = 0.109).

Discussion

During the study, the newly developed camera trap was successfully used to detect and obtain images of percher Sympetrum dragonflies in the field. The percher detector made of passive light sensors was inexpensive and power-saving, as expected. These properties make the detector a major advance for the automated ecological monitoring of perchers.

Of course, extensive analyses, beyond the confirmation of the automatic capture of perchers, are required for the practical application of the camera trap, as is feasible for mammals (e.g., Fukasawa et al., 2016; Nakashima, Fukasawa & Samejima, 2018). For ecological monitoring, data for spatial and temporal dynamics of population density at multiple locations are usually required for statistical analyses. Ideally, perchers should be captured with high accuracy when they land on the detector. Then, the detection frequency can be used as an indicator of population density. It is important to determine whether the detection frequency corresponds to density or is just a rough indicator of presence/absence. Finally, additional traps set across broader areas for longer durations should be demonstrated.

The results of our short-term experiment provide convincing evidence for the performance of the detection device. As expected, the approach was efficient with respect to time. A few tens of false positive detections per day were obtained (consistent with preliminary work in Fukushima and in the field experiment in Tsukuba). This indicated that the type I error rate was acceptably small, considering the time required to check all images obtained in 10-s intervals using a time-lapse camera. Note that commercial camera traps for mammalian monitoring also show a considerable frequency of false-positive detection (Newey et al., 2015). In addition, no organisms other than Sympetrum dragonflies were detected during the short-term field experiment. The sensitivity values for the field experiment over 2 days showed that relatively long perching events were correctly detected. This suggested that the type II error rate was also acceptable.

Furthermore, we observed that the relationship between the detection frequency and dragonfly population density was positive, but not strong. Comparing the camera trap data with investigator data by the goodness-of-fit tests, both methods distinguished whether the dragonfly abundance was low or high (i.e., rooftop vs. other sites). However, different results were obtained for comparisons between the field and the pond, two sites where the dragonfly abundance was relatively high. The GLMM analysis showed that camera detection was predicted from the census data; however, the opposite was not true. This may be explained by the relatively low information obtained by camera detection (15 out of 18 replications were zero values). These analyses suggested that camera detection results were non-linearly related to the population density and that information about population density obtained from a single trap in a short time period is relatively limited compared with census observation data. In our case study, the camera detection results did not adequately reflect the density at the field site. The camera trap at the field site may be less conspicuous for dragonflies due to the adjacent greenhouse. In addition, differences in abundance may have been difficult to detect if a minority of individuals repeatedly occupied the detector section of the camera trap in a location where the dragonfly density was high. If so, the accurate detection of a perching dragonfly does not always indicate population density. The application of the trap to other species with stronger adherence to a territory should be performed with caution.

These results suggested that our simple camera trap may be suitable for distinguishing among locations with high and extremely low dragonfly densities. Of course, there is still room for improvement. In this study, the detection algorithm and parameters were not adequately optimized for Sympetrum dragonflies, despite the relatively high sensitivity, at least in the short-term experiment. An improvement on the rate of false positives owing to movement of the sun and clouds will directly save time for checking captured images. Exploring more sophisticated algorithms may reduce the error and make the tool more convenient. If an improvement in detection accuracy does not correspond to more accurate estimates of population density, better spatial allocation and arrangements should also be explored. Multiple camera traps at a location or multiple detector sections may prevent a minority of individuals from disproportionately occupying the camera trap. In addition, the surrounding environment may affect the behavior of dragonflies. Use in adequately open habitats may be desirable because complex structures will make the camera trap less conspicuous and appealing to the dragonflies. In addition, the application of statistical techniques, such as site occupancy models with heterogeneous detection probabilities (Royle, 2006), will improve population size estimation.

Furthermore, the camera trap only targeted adult dragonflies. To infer the population dynamics of Sympetrum dragonflies more precisely, comparisons with other sampling methods, such as counting exuviae (Tanaka, 2016), are required. Accordingly, data obtained from long-term field experiments with dozens of camera traps are needed.

For the broader application of the trap to percher dragonflies, the ability to accurately identify species is an important issue. There is room for improvement with respect to camera positioning. In the current version of the camera trap, cameras were set about 40 cm below the tip of the detector to prevent dragonflies from perching on the camera, considering the tendency for Sympetrum dragonflies to perch at higher points. This camera positioning provided small images of only the ventral side of individuals. Although the remarkable pattern on the wings can be used to identify the dragonflies in the current positioning, species with key morphologies based on other body parts, such as the thorax, cannot be distinguished. Further research may be required to set the camera appropriately at a higher position and a shorter distance from the detector tip, preventing the camera from interfering with perching behaviors. Connecting a camera with a higher resolution will improve the accuracy of species identification. Considering the recent development of small and power-saving digital cameras, improvements in species identification may be not a major issue for long.

Improvements in hardware may be useful from the viewpoint of sustainability and durability. For example, longer-lasting batteries for the camera will make it possible to use a trap for months. Sealing the electronic circuit with gels, such as silicon, can minimize damage from condensation.

Nevertheless, our camera trap had remarkable advantages over existing sampling methods for percher dragonflies, including its automatic action, low labor requirements, low power consumption, low cost, and non-destructive nature, in addition to the potential improvements and extensions mentioned above. These advantages will be particularly important in areas where access by investigators is restricted (e.g., strict nature reserves and evacuation zones after nuclear power accidents). Furthermore, the camera trap is appropriate for examining the impacts of the abandonment of rice paddy fields on percher dragonflies. Unlike existing sampling methods, such as counting exuviae, camera traps do not require aquatic habitats, which are often lost in abandoned lands (Ikegami, Nishihiro & Washitani, 2011). Of course, combinations with other sampling methods are expected to be effective. For example, the combination with mark and recapture methods (i.e., “recapturing” by camera traps instead of investigators) will save effort for the estimation of population size and dispersal ability.

Conclusions

Our study demonstrated that the ecological monitoring of perching insects using simple and inexpensive camera traps is realistic owing to rapid advancements in digital cameras, processors, storage, and batteries, by implementing a simple strategy based on the behaviors of focal species. Although the scope of our approach in this study is limited to perching insects and not applicable to any others, looking again common behaviors like perching may find a way for automatic monitoring of other type of insects. Our approach is expected to be a key step toward the construction of automatic and well-standardized area-wide monitoring systems for insects and to contribute to global observation networks of biodiversity and the environment.

Supplemental Information

Supplemental Information 1 Circuit diagram for the dragonfly detector

The two CDS resistors (CDS cells) correspond to the two light sensors in the detector section in Fig. 1. The ”Det.” CDS cell and the ”Ref.” CDS cell correspond to the first (upper) sensor and the second sensor, respectively.

Click here for additional data file.

Supplemental Information 2 The C language code implementing algorithm for dragonfly detection

The code was described and compiled using MPLAB IDE v. 8.60 (Microchip Technology Inc.) and microprocessor programming was performed using the MPLAB ICD3 In-Circuit Debugger (Microchip Technology Inc.)

Click here for additional data file.

Supplemental Information 3 The R language code for statistical analyses on dragonfly detection data

The data set was included in the script.

Click here for additional data file.

Supplemental Information 4 The result of camera trapping at each site

Click here for additional data file.

Supplemental Information 5 The perching time of Sympetrum dragonflies at each site obtained by time laps cameras or a video camera

Click here for additional data file.

Supplemental Information 6 The frequency of dragonflies observed at each site

At each site, an investigator counted and recorded the number of Sympetrum dragonflies for 10 min three times a day (six observations in total).

Click here for additional data file.

We are grateful to Dr. M Tamaoki (National Institute for Environmental Studies, Japan) for helping with field work and for providing valuable comments. We also appreciate the valuable and productive comments provided by the handling editor, Dr. Max Lambert, and the two reviewers (one anonymous and Dr. Michael Bogan).

Additional Information and Declarations

Competing Interests

Author Contributions

Patent Disclosures

Data Availability

Our institution (National Institute for Environmental Studies, Japan) supported our research and obtained a national patent (JP 6558701) on the principle idea of perching dragonfly detection using passive light sensors.

Akira Yoshioka, Hiroyuki Oguma, Nao Kumada and Taku Kadoya conceived and designed the experiments, performed the experiments, analyzed the data, prepared figures and/or tables, authored or reviewed drafts of the paper, and approved the final draft.

Akira Shimizu conceived and designed the experiments, authored or reviewed drafts of the paper, and approved the final draft.

Keita Fukasawa conceived and designed the experiments, performed the experiments, analyzed the data, authored or reviewed drafts of the paper, and approved the final draft.

Shoma Jingu performed the experiments, analyzed the data, prepared figures and/or tables, authored or reviewed drafts of the paper, and approved the final draft.

The following patent dependencies were disclosed by the authors:

FLYING ORGANISM DETECTION DEVICE/JP 6558701/26th July 2019: https://jstore.jst.go.jp/nationalPatentDetail.html?pat_id=36004.

The following information was supplied regarding data availability:

The raw measurements and code are available in the Supplementary Files.

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
