# Peer review of "Development of a camera trap for perching dragonflies: a new tool for freshwater environmental assessment"

_PeerJ, doi:10.7717/peerj.9681_

## Round 0.1 · original submission · Major Revisions

Two reviewers have now assessed your manuscript and have found your study to be well-done and generally reported clearly and informatively. Even so, there are a number of revisions that must be met. In particular, both reviewers highlight a number of issues pertaining to methodological (particularly statistical) reporting and clarity.

Thank you for your submission and I look forward to receiving a revised version of your manuscript.

Reviewer 1 ·

Basic reporting

It is my opinion that the manuscript conforms to the standards of reporting in the field.

As I have essentially no electronics expertise, I don’t feel especially qualified to comment on the description of the camera trap itself. However, for what it’s worth, I was able to follow along such that I feel I could replicate their set-up if I wanted to. Perhaps someone with more direct experience with camera traps will feel differently about this section.

The introduction and background provided nice context and cited appropriate literature. The raw data was supplied. I think the content of the figures is nice, though 8 seems kind of excessive for this manuscript and I wonder if there might be ways to combine them.

I have some structural comments that I think will improve the readability of the manuscript, which I elaborate on below, but in general I did not have huge complaints here.

Experimental design

This is original primary research within the scope of the journal.

The research question—can camera traps be used to monitor dragonfly assemblages—is meaningful and relevant. I might personally frame the “hypothesis” that they state on L92-97 differently to reflect the simplicity of this question a bit better, but otherwise I think the question is valid.

I do have some concerns about the rigor of some of the methods, which could simply result from a lack of detail in the methods. I do not think that any of these concerns invalidate the general utility of the camera-trapping method, but I think they may cause the authors to re-consider some of the stated benefits.

1) In my experience doing behavioral surveys of percher dragonflies, it is not uncommon for an individual on a perch to take off and: 1) fight with a rival; 2) patrol its territory; and 3) mate with a female. After this, the dragonfly will often return to the same perch within its territory. Perhaps this is a reflection of the species I have personally studied, but my experience is that oftentimes a single individual will return to a perch within its territory well after 20 seconds has elapsed. As a result, I wonder what their metric of “perching events” is really conveying it seems to me that this could often be the same individual (L231-241). Similarly, if two individuals fight over a territory, and the previous territory-holder loses, then a new individual could plausibly take over the territory and land on the same perch before 20 or even 10 seconds has elapsed. This case wouldn’t register as a new perching event even though it is.

Although none of this will affect the ability of this method to be used for characterizing the presence/absence of a particular species in a particular location, I do wonder about some of the broader inferences the authors seek to make with this kind of method eventually (e.g. L222-226, L304-325). Of course, I feel that even just detecting presence/absence is still very useful and valuable, but certainly it’s also less useful than methods that could eventually be harnessed to provide some estimate of population size. If it is the case that the natural history of the focal species negates these concerns, the authors should state that while also keeping in mind how reliable this method may be for other perchers. One thing that might help is if the authors provide some more detailed information on how sensitive the results are to extending the interval time far beyond 20 seconds (L235-237).

2) The authors take two critical steps to evaluate the robustness of the camera-trapping method, which I appreciated. First, to examine how many “false negatives” the authors were getting from this camera-trapping procedure, I believe (though am not certain) that the authors set up a video camera that continuously monitored the perch to assess how frequently individuals landed on the perch but were not detected by the trap (L216-220). Although I commend the authors for undertaking (what I imagine was) this hugely labor-intensive validation of their camera-trapping method, I felt this section needs considerably more detail. In part, the fact that it is lumped into a paragraph that included many other methods made it not immediately obvious that this was what the authors were describing. Second, the authors also collected census data at their sites to assess if the number of perching events was related to the number of individuals observed by an experienced observer (L222-226). The purpose of this method also wasn’t immediately obvious to me as I was reading it. Moreover, I could not replicate their methods here if I tried. I have a couple of suggestions for improving how these key elements are presented.

a. First, under the subheading CASE STUDY, I would recommend that the authors create a new separate section for the video monitoring procedure and then another separate section for the census. This will help alert readers to the fact that these methods are being conducted, and I think will help the authors describe more clearly about what these methods are for.

b. In the new sub-section for the video monitoring, I’d recommend that the authors begin by stating exactly what this method is for. Then describing the method in detail. Particularly, I would be keen to know more about how the videos were checked thoroughly and matched up to the camera-trapping pictures to get an estimate of false negatives. I would also move the statistical analyses that accompany this validation directly into this section. Here, the authors should provide a clear statement of what prediction is being tested, how it is being tested analytically, and what the readers should expected to see reported if the predictions are supported. Concurrently, I would also put the corresponding results for this into a sub-section in the results with a sub-heading that matches the one used here in the methods. In particular, I would personally be interested in an analysis that considers differences in false negatives among the sites, since it looks like only the pond site had many (Fig. 7).

c. In the new sub-section for the census, I’d recommend that the authors start by providing an explicit rationale for why they did this, as well as what can and cannot be inferred from this method. Then I’d provide a more detailed methodological description of exactly how these 10 min surveys were conducted. For instance, how did the authors move around the site (or did they stand in a single location?). How did the authors ensure that individuals weren’t counted more than once? Was the order in which the sites visited randomized to ensure that site wasn’t confounded with time of day? Then, as with the other new subsection, I’d recommend moving the statistical analyses that accompany these methods into this new subsection. Here again, the authors should provide a clear statement of what prediction is being tested, how it is being tested analytically, and what the readers should expected to see reported if the predictions are supported. Lastly, I would make a corresponding sub-heading in the results to match this new sub-section in the methods.

3) It’s not at all clear to me how the relationship between the detection frequency and observation was analyzed (L278), or what that relationship even is telling us really. I took it to mean the correlation coefficient between the number of perching events at a site and the cumulative frequency of observed dragonflies at that site. If so, n=3 for this analysis, which is not very large for evaluating one of the main stated goals of the paper (L105-107). At any rate, the authors need to clearly state what this test is, what it is for, and report the full statistical output (is this pearson’s or spearnman’s correlation? What is the test statistic with degrees of freedom and the p-value? Etc.).

I also wonder if this is really the best way to look at this relationship given that this small sample size will almost never show a “significant” relationship. I think that a large number of readers will be interested in whether or not camera-trapping could plausibly be used to estimate population size, so I think it would be helpful for the authors to explore the possible benefits and limits as fully as they can. Another way of looking at the data is that the authors have 6 censuses at each of their three sites. If we ignore the non-independence for the moment (we’ll come back to it), there could be 18 total datapoints rather than three. Koenig & Albano (1985, Am. Midl. Natl.) and Moore & Martin (2016, J. Evol. Biol.) found that the average duration over which individuals typically held reproductive territories was ~1.5-2.5 hours, so is there some way to evaluate detection frequencies from the camera-traps in the 1-3 hours around when each census was taken at each site? This would also allow the authors to have match the observations from the camera-trapping method to the in-person observations and perhaps get some estimate of real-time dragonfly activity. To control for the non-independence of site, the authors could consider analyzing this relationship using a mixed-effects model with site as a random effect.

4) L212-215. Please provide more information about the field and pond sites. For example, how large are each of these sites? What vegetation is available for the dragonflies to perch on?

5) L269-271. Is “false positives” a typo here? In the preceding paragraph, the authors say that there were no false positives, and here they say there were many. Since I can’t find the actual number of false negatives anywhere, I wonder if this is what the authors mean? If not, the authors should clarify what they are talking about here, and also directly report the number of false negatives that occurred in text somewhere. It is available in Fig 7, but I think it would be good to have it in text as well.

Validity of the findings

The analyses are mostly appropriate, though, as noted above, could use more description.

Other than the concerns I stated above, I thought the authors did a reasonable job interpreting their results. In future versions of this article, I’d like to see some discussion of how the concerns I raised above influence how the interpretation of data from this camera-trapping technique.

One additional thing I wonder about is that, given that the pictures from the camera-trap are predominantly capturing the ventral side of the dragonfly, what is the broader capacity of this method to discerning among different species (L263-266; figs 2b, 4d)? The authors say they are able to distinguish some species based on wing markings, which is great, but not other species without wing markings. I would like to see some more discussion about the implications of this method if it is difficult to distinguish between most anisopterans.

Additional comments

The submitted manuscript (“Development of a camera trap for perching dragonflies: a new tool for freshwater environmental assessment [#42325]) reports on a novel technique for detecting the presence or absence of perching dragonflies at an area.

I enjoyed reading and thinking about this mostly well-written manuscript. Although I think some revisions are necessary, I believe it could eventually be a nice addition to the literature. In the other sections, I provide my comments on the manuscript following the preferred format of the journal.

·

Basic reporting

This well-written and very interesting manuscript describes a novel apparatus and technique for documenting perching dragonflies. The authors correctly note that this type of technology is quite advanced already for larger animals, but existing methods can’t adequately document insects. I think their device will be extremely useful for dragonfly taxa that perch—in their study, they focus on Sympetrum, but there are many other perching dragonflies that could be detected with this approach. I think this manuscript will make a valuable contribution to the study of larger insects globally. Most of the figures are very useful to the manuscript, but below I provide some specific feedback for figures 6-8.

Experimental design

The authors are to be commended for explaining the technical set-up as simply as possible for a general audience. Although I have no technical expertise in the types of electronics used in the study, I was mostly able to follow their descriptions. In addition to designing, building, and testing the apparatus, the authors deploy the apparatus in three different settings and use a secondary time-lapse camera as well as expert observers (dragonfly ecologists) to test the sensitivity of the device (e.g. false positives and false negatives)—which is an excellent approach.

Validity of the findings

In general, the authors do a good job providing a proof-of-concept for the apparatus. However, I’m a little confused about the statistical approach of this manuscript. Why is the main question of statistical interest whether the “probability of dragonfly occurrence is equal between sites” (line 246)? The authors test their apparatus at three sites-- a pond, an open field, and a building rooftop—which are very dissimilar to one another and in what resources they could provide for dragonflies. What’s the ecological reasoning for assuming an equal distribution of dragonflies among these three sites? I would assume from the start that there should be an unequal distribution across these sites—because the rooftop would not have much to offer in terms of food or habitat. And why is that question of interest for the proof-of-concept for the apparatus? It seems like the real question here is if the efficiency or accuracy of the device differs in these three settings (e.g. number of false negatives relative to true positives; or the frequency of false positives compared to true positives) -- but it doesn’t seem like that’s what this manuscript’s statistics address. I would like to see some additional clarity and justification for the authors’ current approach if it is to remain, and would suggest they think deeply about what they’re asking and why—and perhaps adopt a different statistical approach based on that. Finally, it would be valuable to statistically compare the densities/frequencies of dragonfly sightings from the human observers to those of the apparatus— relative comparisons are made in the Discussion section, but this could be addressed quantitatively too.

Additional comments

Minor comments:
Line 161—“a continuous perching signal for Z seconds was required” – what do the authors mean here by ‘Z seconds’? It becomes apparent later in the paper, but should be explained better here.

Line 225—was the cumulative number of dragonflies observed really used as the “frequency of dragonflies observed”? It seems that this should be expressed as a unit of abundance over time, e.g. # of dragonflies observed per minute or per hour.

Line 254—“Add your materials and methods here.” ??

Figure 6 through 8-- formatting looks unprofessional (e.g. inconsistent fonts types and sizes)

Figure 8—make the y-axes the same, need more info in caption to make it standalone

Line 288—citations?

Line 297—why is a value of “a few tens of false positives” considered “very low”? Especially considering the relatively low number of true positives (e.g. only 50 true positives at the pond site)

---

## Round 0.2 · Minor Revisions

Both reviewers have now closely evaluated your revised manuscript and are excited by the improvements. I thank the authors for their careful attention to the reviewers' comments. The reviewers only have a few minor details that need addressing, predominantly with respect to clarity and word choice. These minor revisions will improve the accuracy of the manuscript and should be fairly straight-forward to accomplish.

Reviewer 1 ·

Basic reporting

I continue to be pleased with the basic reporting of this manuscript and believe that it adheres to standards of this journal.

Experimental design

The authors have addressed all of my previous concerns for this section, and I believe the manuscript meets the standards of the journal.

Validity of the findings

The authors have addressed all of my previous concerns for this section as well, and I believe the manuscript meets the standards of the journal.

Additional comments

I commend the authors for their substantial revision of the manuscript, and I look forward to seeing this manuscript published soon. I have only a few (very minor) outstanding comments about wording that the authors might consider prior to publication.

L231-232. "For photos obtained within a few seconds, the first was removed from the analysis". I think a word must be missing here. This is for instances where there are multiple photos within a few seconds, correct?

L337-349. These two sections report on different components of the overall project, and I think it would be helpful if the first sentence of each section re-stated what the ensuing results are in reference to. For instance, L342 starts by immediately describing the number of true positives in the field study. However, the sentence that immediately preceded it (in the previous section) described the number of false positives. As I was reading this, I initially reacted by adding those numbers together in my head and thought that the number of false positives vastly exceeded the number of true positives. Having a sentence at the beginning of each section that just briefly restates what the goal of each section is will help readers remember that the results refer to totally different contexts.

L397-398. "The GLMM analysis showed that camera detection was related to the population density, as measured by the census data; however, the opposite was not true." To clarify the directional nature of these statistical relationships, I would recommend changing this sentence to "The GLMM analysis showed that camera detection predicted population density, as measured by the census data; however, the opposite was not true."

·

Basic reporting

This revised article represents a good improvement over the first iteration.

Experimental design

I appreciate the work that the authors have done to improve the clarity of the statistical approaches in this revision. I'm still a little uncertain as to why the goodness-of-fit tests are included to compared differences in detections among sites, since there is no reason to think that these three sites (field, pond, rooftop) would have similar numbers of dragonflies anyway, but that's okay. The authors clearly describe the patterns of false positives, and the relationship between the trained observer and the dragonfly camera (with the new models they added), so the important aspects are covered well. And the lack of false negatives in the entire study is very encouraging for the approach.

Validity of the findings

For the most part, the authors do a good job talking about the limitations of the study, and addressed well the concerns that both myself and the other reviewer had about the previous version of this manuscript. My final suggestions would be to:

(1) be careful when talking about the future uses of this approach. For example, in line 457, the word "perching" needs to be inserted before "insects". This approach is very promising and exciting, but it only applies to perching insects and not any others. This is highlighted by the fact that the rooftop site did indeed have dragonflies (I think 8 were observed in flight by the human observer), but none were detected by the camera because none perched. So be careful here and throughout to qualify the approaches use with "perching".

(2) highlight the issue with 'false positives' a bit more in the discussion when talking about limitations of this novel and exciting approach. The rate of false positives due to sun or clouds was pretty high at times, so it seems like a goal for follow up studies should be to reduce that rate of false positives before trying to use this approach widely.

Additional comments

I am excited to see this technology and approach develop further to enhance our understanding of perching dragonflies and other insects!

---

## Round 0.3 · accepted · Accept

Thank you for your careful and thorough responses to the reviewers in revising your manuscript and congratulations!